# Integrating a brief mental health intervention into primary care services for patients with HIV and diabetes in South Africa: study protocol for a trial-based economic evaluation

Vimbayi Mutyambizi-Mafunda,[1] Bronwyn Myers,[2] Katherine Sorsdahl,[3] Crick Lund,[3,4] Tracey Naledi,[5,6] Susan Cleary[7]

For numbered affiliations see end of article.

**Correspondence to**
Ms. Vimbayi Mutyambizi-Mafunda;
vimbayi.mafunda@uct.ac.za

## ABSTRACT

**Introduction** Depression and alcohol use disorders are international public health priorities for which there is a substantial treatment gap. Brief mental health interventions delivered by lay health workers in primary care services may reduce this gap. There is limited economic evidence assessing the cost-effectiveness of such interventions in low-income and middle-income countries. This paper describes the proposed economic evaluation of a health systems intervention testing the effectiveness, cost-effectiveness and cost-utility of two task-sharing approaches to integrating services for common mental disorders with HIV and diabetes primary care services.

**Methods and analysis** This evaluation will be conducted as part of a three-armed cluster randomised controlled trial of clinical effectiveness. Trial clinical outcome measures will include primary outcomes for risk of depression and alcohol use, and secondary outcomes for risk of chronic disease (HIV and diabetes) treatment failure. The cost-effectiveness analysis will evaluate cost per unit change in Alcohol Use Disorder Identification Test and Centre for Epidemiological Studies scale on Depression scores as well as cost per unit change in HIV RNA viral load and haemoglobin A1c, producing results of provider and patient cost per patient year for each study arm and chronic disease. The cost utility analyses will provide results of cost per quality-adjusted life year gained. Additional analyses relevant for implementation including budget impact analyses will be conducted to inform the development of a business case for scaling up the country's investment in mental health services.

**Ethics and dissemination** The Western Cape Department of Health (WCDoH) (WC2016_RP6_9), the South African Medical Research Council (EC 004-2/2015), the University of Cape Town (089/2015) and Oxford University (OxTREC 2–17) provided ethical approval for this study. Results dissemination will include policy briefs, social media, peer-reviewed papers, a policy dialogue workshop and press briefings.

**Trial registration number** PACTR201610001825405.

## Strengths and limitations of this study

► This study will provide some of the first empirical evidence of the cost-effectiveness of integrating services for common mental disorders in the primary care offering for HIV and diabetes.

► The study will generate findings that may guide priority setting through application of quality-adjusted life year-based outcomes and decision-making around the implementation of mental health counselling within the health system.

► This paper adds to the growing body of protocols for economic evaluations conducted alongside randomised controlled trials.

► As health service use is self-reported, there is a possibility for recall bias, to limit this, a 1-month recall period will be used for ambulatory services and a 6-month recall for hospitalisation, and hospitalisations will be validated through the Department of Health Data Centre.

► The 1-year follow-up period may reflect outcomes that do not fully represent the full benefit that patients and the health system could gain through the availability of this service; further work using an extended follow-up period may be proposed.

## INTRODUCTION

Common mental disorders (CMDs) including depression, anxiety and alcohol use disorders are highly prevalent conditions that impose a significant societal cost and impact on quality of life.[1] Globally, these conditions have been shown to be the leading cause of non-fatal burden of disease.[1 2] The disabling impact of these conditions is particularly significant in patients suffering from chronic physical conditions.[3] CMD comorbidity has been shown to exacerbate major modifiable risk factors for chronic disease; contribute to chronic disease progression; increase the prevalence of preventable complications and

disability; result in poor adherence and ultimately lead to treatment failure.[1 4] There is a plethora of evidence on the impact of depression and alcohol use disorders on diabetes outcomes.[5 6] Likewise, alcohol consumption and depressive conditions have been associated with poor HIV outcomes.[7 8] In addition, high-income country (HIC) evidence suggests that patients with these multimorbidities have higher healthcare costs even after adjusting for the costs of mental healthcare[9]; suggesting potential cost savings from investing in mental health treatment for patients with multimorbidities.[10]

There is a pressing need for evidence on cost-effective strategies for integrating mental healthcare into primary health services, particularly in low-income and middle-income countries (LMICs), where health systems are typically fragile and chronically underfunded. In these health systems, an investment in mental health services at primary care level for patients with chronic disease has the potential to result in significant returns on investment due to reduced chronic disease treatment costs. South Africa—with the world's largest ART programme,[11] an increasing prevalence of diabetes[11] and a huge unmet need for CMD treatment[12]—is one LMIC that may benefit from this integrated approach.

Task-sharing[13 14] involves redistributing tasks from specialist providers to non-specialist health workers,[15] with specialists providing ongoing support to these health workers. Task-sharing is a common strategy for the provision of primary care services in constrained LMIC settings.[16] Due to the long lead times in training mental health staff and financial resource limitations, the task-sharing discourse has found appeal in the policy discussions around introducing and expanding mental healthcare in LMICs.[1 17] These discussions are centred on using task-shared approaches[18] to upskill lay health workers to integrate mental health services such as brief counselling interventions into primary care.[19 20] Task-sharing the psychosocial counselling of patients who have chronic comorbidities to lay health workers is a particularly appealing and potentially cost-effective strategy to treat patients and support their adherence to chronic medication[21 22] and therefore limit overall health system costs of care.[4 16]

While task-sharing the treatment for CMDs from mental health professionals to lower level healthcare workers has been suggested as an effective way to deliver services within the context of scarce human resources[4] and has been shown to be cost-effective in some HIC settings, there is little if any economic evidence to support the implementation of such strategies in LMICs.[4 16] This lack of evidence limits decision making around the allocation of resources for these services. Economic evidence on the institutional investment, cost-effectiveness and budget impact of this approach is necessary for empirically informed health service planning.[23] This work presents a protocol of a cost-effectiveness (CEA) and cost-utility analysis (CUA) from a societal perspective (ie, providers and patients) of a task-shared mental health counselling intervention for patients with comorbid HIV or diabetes in South Africa.

## METHODS AND ANALYSIS
### Patient and public involvement
Patients and the public were engaged with through the trial, and were not directly involved in this study.

### Trial design
This study is nested in Project MIND, a cluster randomised controlled trial across 24 public primary care facilities.[24] Myers *et al*[24] describe the trial in detail. The trial will compare the effectiveness of treatment as usual (MIND_TAU) and two alternative methods for integrating a psychosocial intervention into the primary care services for HIV and diabetes. Treatment as usual (TAU) is representative of typical primary care level services for CMD in the South African public health service facilities, which is typically limited to referrals.[25 26] The two models will be the dedicated and designated models of care. In the dedicated approach (MIND_DED), community health workers (CHWs) will be hired and added to the facility staff complement and will dedicate their time to *only* delivering the new counselling service. In the designated approach (MIND_DES), CHWs already working in the facility will be designated to deliver the service *in addition* to their other chronic disease-related activities such as adherence counselling for HIV and health promotion. Supervision and debriefing of all the counsellors in the active arms will be task-shifted to registered psychological counsellors in line with the national mental health policy framework.[27] Descriptions of the CHWs and registered psychological counsellors roles in the MIND-DED and MIND-DES models and their qualifications and skills levels are detailed in the trial protocol.[24]

### Intervention and comparator
The counselling programme delivered in MIND-DED and MIND-DES is based on motivational interviewing (MI) and problem solving therapy (PST).[28] It is delivered in three sessions over 6 weeks with one optional booster session. The MI component provides psychoeducation about depression and hazardous alcohol use in relation to chronic disease, builds readiness for counselling and helps participants develop goals for counselling. The PST component targets maladaptive coping strategies and builds skills and strategies for dealing with life problems. Patients will receive individual face-to-face counselling from CHWs and take-home booklets to support self-learning between sessions. Registered psychological counsellors, supervised by psychologists, will train the counsellors.[24] The underlying theory, content and evidence for the efficacy of the counselling approach for reducing symptoms of depression and hazardous/harmful alcohol use over the 1-year time frame have been described previously.[29–31] In the TAU arm, the standard package of care will be provided to patients who are

suspected of having mental health problems. In general, patients using the HIV or diabetes service are asked by a nurse or doctor attending to their care about their mental well-being, life stressors and use of alcohol or other substances. Patients are provided with advice to change their lifestyles. Where the healthcare provider deems it necessary, patients are referred to a mental health nurse for further assessment or screening. The patient may also be referred to a social worker who may refer them to non-governmental organisations (NGOs) who provide counselling and support services.[24]

## Outcome measures

The primary outcomes for the trial are measures of psychosocial functioning, specifically changes in self-reported risk of depression as measured by the Centre for Epidemiological Studies scale on Depression (CES-D)[32] and self-reported hazardous alcohol use as measured by the Alcohol Use Disorder Identification Test (AUDIT).[33] Secondary outcomes include biological measures aimed at assessing changes in chronic disease treatment response, specifically haemoglobin A1c (HbA1c) for diabetes and HIV-1 RNA viral loads for HIV. Other secondary outcome measures include self-reported adherence to chronic disease medication and health-related quality of life (HRQoL) measured using the EuroQol-5D (EQ-5D).[34] All outcomes are measured longitudinally at baseline, and at 6 and 12 months follow-up assessments, recruitment on the trial started in 2017 and will end in February 2019 and final outcome assessments will end in 2020 (figure 1).

## Sample size and patient population

Recruitment will be restricted to consenting adult patients who meet criteria for hazardous alcohol use or depression using the AUDIT or CES-D. These patients are already diagnosed and receiving treatment for diabetes or HIV, but not currently receiving treatment for a CMD. The trial requires a sample size of 1200 patients (600 with HIV and 600 with diabetes) to detect reductions in hazardous alcohol use and risk of depression at 12-month follow-up. The sample size calculations are based on a cluster randomised design with two active arms and a control arm. This clinical trial is powered to detect clinical outcomes, specifically reductions hazardous/harmful alcohol use and risk of depression at 12-month follow-up rather than economic outcomes.[24 35]

## MIND economic evaluation
### Objective

The study will be a prospective economic evaluation. The objectives include estimating:

i. Full economic provider costs of the mental health intervention; any cost offsets attributable to the intervention related to changes to the costs of HIV or diabetes care at the primary care level and changes to the costs of referral care (including tuberculosis (TB), emergency department and inpatient care).

ii. Patient costs: associated with the intervention which will include direct out-of-pocket payments to private health providers (consultations and medications, etc), travel and subsistence costs and productivity losses.

| Analysis | Assessment/ Economic Evaluation Outcomes | Measurement instrument OR Clinical Parameter | Assessment Timing (Trial recruitment: starts 2017- ends 2019) | | | |
|---|---|---|---|---|---|---|
| | | Intervention start | Baseline | 6 months | 1 Year | Intervention assessment end |
| **CEA** | *1° Outcomes* | | | | | |
| | Harmful alcohol use (Self-report interview) | AUDIT | ✓ | ✓ | ✓ | |
| | Depression (Self-report interview) | CES-D | ✓ | ✓ | ✓ | |
| | *2° Outcomes* | | | | | |
| | Diabetes treatment failure risk | HbA1c | ✓ | ✓ | ✓ | |
| | | BMI | | | | |
| | HIV treatment failure risk | HIV-1 RNA (HIV-viral load) | ✓ | ✓ | ✓ | |
| | Chronic Disease Medication Adherence | Visual analogue Scale and CASE adherence Index | ✓ | ✓ | ✓ | |
| **CUA** | *Multi-attribute Outcomes* | | | | | |
| | QALY | EQ-5D | ✓ | ✓ | ✓ | |

**Figure 1** Economic evaluation: analyses, outcomes, measurement and assessment timing. AUDIT, Alcohol Use Disorder Identification Test; BMI, body mass index; CEA, cost-effectiveness analysis; CES-D, Centre for Epidemiological Studies scale on Depression; CUA, cost- utility analysis; EQ-5D, EuroQol-5D; HbA1c, haemoglobin A1c; QALY, quality-adjusted life year.

iii. Cost-effectiveness in terms of the incremental cost per unit of improvement in disease-specific outcome (CES-D, AUDIT, HIV-1 RNA, HbA1c).

iv. Cost-utility in terms of the incremental cost per quality-adjusted life year (QALY) gained.

v. Budget impact of scaling up the intervention.

To achieve these objectives, the incremental cost-effectiveness ratio (ICER) will be calculated by ordering the MIND_DED, the MIND_DES and MIND_TAU alternatives from least cost to highest cost and calculating the difference in costs and outcomes against the next less costly, non-dominated alternative or against the common baseline of MIND_TAU.

## Perspective

The aim of the economic evaluation is to inform health service decision makers about the costs, outcomes, economic efficiencies and budget impact of introducing task-shared psychosocial counselling to their chronic disease patient body in a resource-scarce context. As such, the economic evaluation will be conducted from a provider perspective, reflecting health service budget constraints. In line with good practice recommendations, the analyses will also be presented from a societal perspective (including both provider and patient perspectives). While public sector primary healthcare is free at the point of use, patients incur time and travel costs when accessing care and may experience productivity losses; in addition, they may incur costs when using private sector health services. These costs will be collected to inform the patient perspective within the economic evaluation.

## Time horizon and discount rate

As the health consequences of the intervention will be experienced over a period extending longer than 1 year, a discount rate will be applied to reflect differentials in timing.[36] A 3% discount rate will be used for discounting both costs and outcomes to allow comparability with other CEAs.[37]

## Estimating intervention effects

Single study-based estimates of clinical effectiveness will be obtained from the trial, which will provide the primary and secondary outcome measures previously described. In addition to the disease-specific primary outcomes, and the secondary HIV and diabetes outcomes which will be applied in the CEA, the QALY will also be calculated as an end point that will be used in the CUA. The ICER resulting when using a QALY measure represents the additional cost associated with accruing another QALY as one compares strategies (ie, the incremental cost per QALY gained). The QALY allows comparison of ICERs across different diseases and interventions.[37] QALYs are calculated by multiplying the length of time in a health state by the HRQoL of the health state. The EQ-5D is the most widely used instrument for measuring HRQoL for CUAs.[34] In South Africa, it has been validated and translated into Afrikaans and isiXhosa, two of the country's official languages.[38] The EQ-5D is a self-assessment of health status across five domains which include mobility, self-care, usual activities, pain and discomfort and anxiety/depression.[39] In this study, the EQ-5D-3L[34] will be used to measure HRQoL. A single summary index score will be generated using preference weights derived from a time trade-off (TTO)-based valuation survey conducted in Zimbabwe in addition to the commonly used UK value set; the TTO is the most common valuation technique used in LMICs.[40] As the EQ-5D will be administered at all the assessment intervals (baseline, 6 months and 12 months post-enrolment), there will be three observations of the EQ-5D scores for each patient in the trial. The QALY gained per patient over the trial period will be calculated by measuring the area under the quality of life curve.[37]

## Estimating costs for the provider's perspective

Within economic evaluation, the appropriate scope of provider costs includes all costs incurred within the intervention and any changes in broader health system costs that can be attributable to the intervention. We have categorised these costs as intervention costs, HIV and diabetes service costs and other related provider costs. As shown in table 1, our approach to estimating these costs entails the measurement of quantities of resources that are used, and multiplying these quantities by the value (or unit cost) of each resource. These separate steps of measurement and valuation are described below.

## Measurement of resource use

For the intervention costs, an ingredients approach will be used to estimate resources. Routine data linked to intervention protocols will be collected and used to assess resources directly consumed in the provision of the intervention, including supplies, manuals and patient education materials. Facility observations of intervention delivery will be conducted for the economist to understand the intervention and the facility context; these observations will complement intervention cost data collection. The costs of intervention focused counsellor training and facility-level institutional strengthening and capacity development through organisational readiness workshops[41] will be included in the cost analysis as start-up costs[42] in order to capture the full economic costs of integrating the intervention into the primary care service for patients with chronic disease.[42]

A time and motion tool (TMT) developed for this evaluation will measure counsellor time usage in delivering the intervention for both active arms in the study. Time tools have been used in analysis of human resource requirements for expanding access to HIV[43] and TB care[44] using lay health workers and have proved useful in highlighting the scope of duties of this cadre. To our knowledge, there is limited evidence in the South African context on time costs of CHW-delivered mental health counselling. The TMT will enable an analysis of differences in counsellor time usage in the different intervention modalities,

**Table 1**  Measuring and valuing provider costs

| | Measurement | | Valuation | |
|---|---|---|---|---|
| **Cost component** | **Resources used** | **Quantities or utilisation** | **Valuation data** | **Allocation factor** |
| Intervention | | | | |
| Capital costs | | | | |
| Facility readiness workshops | Number of staff | Time from trial data | Cost of employment converted to annual equivalent cost | DED/DES/TAU headcount |
| Intervention training for counsellors | Number of counsellors | Time from trial data | Cost of employment converted to annual equivalent cost | DED/DES/TAU headcount |
| Counselling room | Space | Square metres | Replacement value converted to annual equivalent cost | DED/DES/TAU headcount |
| Furniture and equipment | Tape recorders | Inventory | Replacement value converted to annual equivalent cost | DED/DES/TAU headcount |
| Vehicles | Vehicles | Inventory | Replacement value converted to annual equivalent cost | DED/DES/TAU headcount |
| Recurrent costs | | | | |
| Counselling personnel | Number of staff (lay counsellors; registered counsellors; clinical psychologist) | Time from time and motion tool | Cost of employment×proportion of time | DED/DES/TAU headcount |
| Counselling supplies | Manuals, notebooks, pens | Number used—from trial data | Market value | DED/DES/TAU headcount |
| Non-intervention personnel | Number of staff | Time from facility data | Cost of employment×proportion of time | Facility headcount |
| Utilities | Electricity, water, other utilities, phone, cleaning, transport and stationery | Facility utilisation | Annual facility expenditure | Facility headcount |
| HIV/Diabetes service | | | | |
| Capital costs | | | | |
| Buildings | Space used for HIV/ diabetes service | Square meters | Replacement value converted to annual equivalent cost | HIV/Diabetes headcount |
| Equipment and furniture | Equipment and furniture used for HIV/diabetes service | Inventory | Replacement value converted to annual equivalent cost | HIV/Diabetes headcount |
| Vehicles | Vehicles used for HIV/ diabetes service | Inventory | Replacement value converted to annual equivalent cost | HIV/Diabetes headcount |
| Recurrent costs | | | | |
| HIV/Diabetes personnel | Number of staff and staff time spent on HIV/diabetes service | Time from facility data | Cost of employment ×proportion of time | HIV/Diabetes headcount |
| Drugs | ARV/Diabetes medication | Treatment protocol | National tender prices | Direct allocation |
| Diagnostic tests | CD4 test, diabetes monitoring tests, etc | Treatment protocol | National Health Laboratory Service prices | Direct allocation |
| Non-HIV/diabetes personnel | Managerial, cleaning and security staff | Facility utilisation | Facility expenditure | Facility headcount |

Continued

**Table 1** Continued

| Cost component | Measurement | | Valuation | |
| | Resources used | Quantities or utilisation | Valuation data | Allocation factor |
| --- | --- | --- | --- | --- |
| Building operating and maintenance | Electricity, water, other utilities, phone, cleaning, transport and stationery | Facility utilisation | Facility expenditure | Facility headcount |
| Other related healthcare providers | | | | |
| Public clinic visit | Average cost per visit | Patient interviews | Facility expenditure | Facility headcount |
| Public hospital ER/OPD visit | Average cost per OPD visit | Patient interviews | Facility expenditure | Facility Inpatient days and Outpatient headcount |
| Public hospital inpatient day | Average cost per inpatient day | Patient interviews | Facility expenditure | Facility Inpatient days and Outpatient headcount |

indicating opportunities for scale and scope efficiencies. It is anticipated that the costs associated with the TAU option will be minimal and any referral costs will be captured through patient self-report of services they have used.

For the HIV and diabetes services, a step-down methodology will be used to establish resource use. Staff time will be determined through review of facility organograms and interviews with senior managers. In addition, utilisation of HIV and diabetes medicine and diagnostics will be estimated by applying standard treatment guidelines/protocols for HIV and diabetes treatment in primary care services to utilisation statistics obtained from facility records (ie, protocol-based facility utilisation statistics)

Establishing the use of referral services is challenging given that there is no health information system that enables patient pathways through care to be established. Therefore, to establish patient utilisation of health services beyond their HIV and diabetes care at the primary level, we will administer a patient questionnaire developed for the study at baseline, and at the 6-month and 12-month assessments. Using this questionnaire, we will capture health service utilisation for other primary care services (eg, TB treatment), hospital services, emergency department care, paramedic care and referrals to mental health providers. Patients will be asked to quantify the number of these additional visits, and lengths of stay in hospital. Given concerns with patient recall, we have asked patients to estimate their usage of ambulatory services over the previous month and have used a recall period of 6 months for inpatient services. These data will then be extrapolated to estimate costs over the time horizon.

### Valuation of costs
Once the utilisation of services has been established, we will estimate the unit cost of each service. This involves sourcing a value for each item (eg, annual cost of employment multiplied by proportion of time spent), as well as an allocation factor (eg, annual facility head count) allowing the estimation of a unit cost (cost of

employment×proportion of time/headcount). These sources of data and allocation factors are outlined in detail in table 1.

For capital items, costs will be estimated using the replacement value method.[37] In essence, the current replacement value of each items is calculated and annuitized using data on the estimated useful life of the item and an interest rate representing the opportunity cost component for capital, in order to estimate an equivalent annual cost. An estimated useful life of between 5– and 20 years for furniture and buildings will be applied as while an interest rate of 8% will be used as the annuitization factor based on the rate of return on government bonds in South Africa.[37]

### Estimating costs for the patient perspective
#### Measurement of resource use
For the patient perspective, we will use a questionnaire at baseline, 6 and 12 months to estimate resource use. This will capture any user fees associated with public, private or NGO-provided health services as well as traditional and faith-based therapies. In addition, we will ask patients to estimate their time spent accessing healthcare services, as well as out-of-pocket payments for transport, subsistence and accommodation. Travel, subsistence and accommodation costs will relate to both the patient and their caregiver, if the patient is accompanied by a caregiver to any of these services.

#### Valuation of costs
Travel, subsistence and accommodation costs will be estimated using the out-of-pocket payments reported by patients. Time spent receiving healthcare will be valued using the equality of wages approach.[45 46] We will apply this method as public health services in South Africa are typically accessed by those who have a lower socioeconomic status. Using this technique, the time costs for all patients are valued at the average income reported by the survey population.[45] This approach is expected to provide a relatively accurate reflection of the opportunity cost of

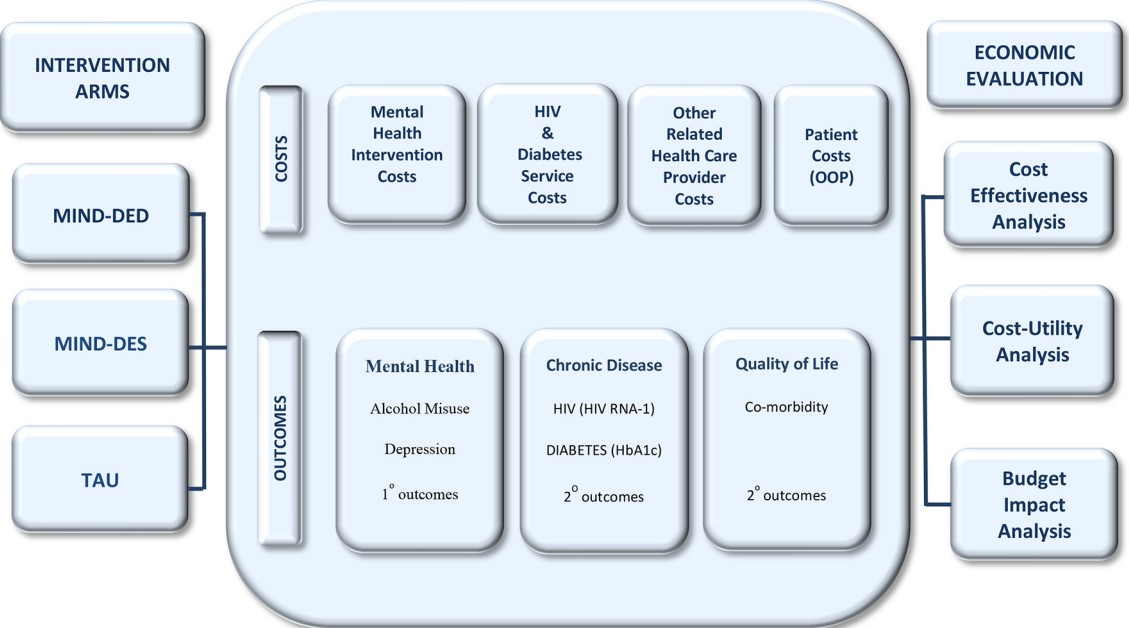

**Figure 2** Project MIND: intervention arms, costs, outcomes and economic evaluation. HbA1c, haemoglobin A1c; OOP, out-of-pocket; TAU, treatment as usual.

unpaid labour and is favoured over the minimum wage approach in this context as a large proportion of the patients are unemployed resulting in a possible overestimate of the value of the time loss if the minimum wage is used.[45] In addition, we will ask patients if they have lost any income through seeking healthcare.

Figure 2 summarises the intervention arms, costs and outcomes that will be included in this study.

All costs will be valued at 2017/2018 prices and, to allow comparability with other studies will be converted from South African Rand (ZAR) to US$, using appropriate conversion rates for the same period obtained from OANDA, an online currency exchange rate conversion site. Costs incurred during earlier periods (such as manual development) will be inflated using the Consumer Price Index to reflect 2017/2018 prices.

### Analyses
#### Statistical analysis of costs and effects
Mean provider costs and patient costs will be calculated from baseline until the end of the intervention period. In addition, we will explore the extrapolation of costs to later periods if data allow for this. Cost differences between the MIND-TAU, and the MIND-DES and MIND-DED alternatives will be calculated. Sampling uncertainty for cost data will be estimated using statistical methods that account for the non-normal distribution of costs. Effect in terms of CES-D, AUDIT, HIV viral load, HBA1c and the QALY will be analysed using linear regression on intervention type.

#### Determining cost-effectiveness
The analysis will be conducted in line with the Consolidated Health Economic Evaluation Reporting Standards guidelines.[47] Decision analytical models[37] integrating the cost and outcomes of the different intervention alternatives will be developed to assess cost-effectiveness. As a meaningful impact of the MIND-DES and MIND-DED on both costs and effects is anticipated, ICERs will be estimated in terms of incremental cost per natural unit and/or intermediate outcome for the CEA, and incremental cost per QALY gained for the CUA. As the trial is focused on health systems strengthening for the delivery of mental health services, the decision rule applies to maximising health within the healthcare budget constraint. Therefore, ICERs will be calculated from the provider perspective and, the cost per QALY gained will be compared with the cost-effectiveness thresholds (CETs) for LMIC settings.[48] If our intervention's ICER is less than the chosen CET, this will mean that diverting resources to the intervention will increase population health, and if the ICER is more than the CET the intervention is not cost-effective. Patient costs will be collected and reported on separately. In addition, to satisfy the requirements of common guidelines[47] and to allow for some degree of comparability with other studies, the ICERs will also be presented for the societal perspective, that is, including both provider and patient costs.

#### Sensitivity analyses
The decision analytical models will allow for univariate and multivariate sensitivity analyses to be conducted. This will enable the robustness of study findings to be tested by varying key parameters such as utilisation rates and assessing the impact on cost-effectiveness results. Standard areas of analytical focus will be considered, including testing sensitivity around the resource cost and prices, and theoretical controversies,[49] for example, TTO weights[50] and discounting of outcomes.[36] Further

**Table 2** Planned sensitivity analyses

| | Base case | Range |
|---|---|---|
| **Simple sensitivity analysis** | | |
| Discount rate | 3% | 0%–10% |
| QALYs: HRQoL weights | EQ-5D valued using UK TTO | EQ-5D valued using Zimbabwe TTO |
| Unit costs | Primary data and data from published sources and official statistics for South Africa | Varied within plausible ranges as determined from a literature review of South African cost studies |
| Missing data | Multiple imputation | Complete case analysis |
| **Probabilistic sensitivity analysis** | | |
| Intervention utilisation data | Mean value | 95% uncertainty interval |
| Referral service utilisation data | Mean value | 95% uncertainty interval |
| EQ-5D/CES-D/AUDIT data | Mean value | 95% uncertainty interval |
| Clinical outcomes (HbA1c, HIV-1 RNA) | Mean value | 95% uncertainty interval |

AUDIT, Alcohol Use Disorder Identification Test; CES-D, Centre for Epidemiological Studies scale on Depression; EQ-5D, EuroQol-5D; HbA1c, haemoglobin A1c; HRQoL, health-related quality of life; QALY, quality-adjusted life year; TTO, time trade-off.

analyses to inform the economics of task-sharing will include sensitivity analysis focused on economies and diseconomies of scale and scope, using detailed data from the TMT.[51] This will include testing variations in staff hours worked, variations in staff mix, lay counsellor funding models and infrastructure investments. The impact of intervention dosage on outcomes will also be assessed in the sensitivity analyses. Statistical methods including multiple imputation will be used to manage the effect of missing information on costs and effects.[52] These sensitivity analyses are summarised in table 2. Probabilistic sensitivity analysis (which allows application of CIs to point estimates) will be used for sensitivity analyses of all relevant individual-level variables. The sensitivity analyses will be used to produce a cost-effectiveness acceptability curve.

## Budget impact analysis

There is an increased demand from decision makers in both LMIC and HIC settings for estimates of the real fiscal implications of investing in health services within a defined budget and budgetary period.[53 54] Trial-based estimates of intervention uptake and population-based estimates of numbers in need of the intervention will be used in budget impact approximations,[55] which will be conducted using a mathematical programming approach.[56 57] Mathematical programming provides a useful equity lens for informing policy decisions around the provision of care in high burden, high cost contexts.[58] It will involve analysing alternative implementation strategies with differing equity implications subject to a budget constraint. This approach aims to support policy making by applying principles of constrained maximisation in resource allocation.[58] The estimates from this study will inform policy-level discussions on the fiscal implications of investing in primary care mental health services.

## ETHICS AND DISSEMINATION

In the CEA we foresee minimal risks to participants, as patient level data will be collected through the main trial, where, in order to minimise the risk of improper disclosure of information study staff will be required to sign a staff confidentiality agreement and will be trained and certified in protecting human participants. Data collection for the trial will be conducted using Computer-Assisted Personal Interviewing (CAPI) in which only a unique study identifier is used. Knowledge translation will be through policy briefs, and social media as well as peer-reviewed papers, and presentations at health economics and mental health policy conferences. Engagement with department of health policy makers and other key stakeholders will be through a policy dialogue workshop linked to the main trial's Stake Holder Advisory Group. Dissemination of results to the public will be through press briefings in national and local media.

## DISCUSSION

The delivery of psychotherapy using varying models of task-sharing to less specialised health workers in primary care has been shown to be cost-effective in HIC settings across a range of mental health conditions[59–61] but has limited empirical economic evidence in LMICs.[51 62 63] This study will provide evidence to address this gap. Through the CEA, the relative efficiency of each delivery mechanism will be established, enabling a discussion around which model produces the best outcomes given the available resources. As staff time is expected to be one of the main drivers of costs due to the nature of the intervention, we have developed a time tool that will allow us to directly measure staff time usage and investigate any potential excess capacity (hypothesized to occur with the MIND-DED counsellors), and scope economies

(hypothesized to occur with the MIND-DES counsellors), which would impact the comparative efficiency of the different alternatives and thus the cost-effectiveness results, as suggested in a similar study.[51] The CUA will aid health service planners in their decision making around resource allocation and may be useful for guiding priority setting decisions both within the mental health budget space and across competing health priorities. To further support policy maker decisions, the affordability of implementing the intervention will need to be assessed, this will require further analysis using trial-based evidence on intervention uptake and estimates of population-level numbers in need to approximate the budget impact of equitably implementing the service.[58]

Policy discussions on the integration of services to provide holistic care for patients requires economic evidence to support implementation in resource-limited contexts.[4 23] By estimating the full economic costs of integrating the mental health treatment into the primary care offering for patients with chronic physical diseases, this study will provide much needed economic evidence to support this policy narrative.[4 64 65] Empirical data on the costs of upskilling and training staff to deliver the intervention and the time costs of the alternative human resourcing strategies will also aid in understanding how this cadre of staff organise their work and may highlight opportunities for efficiency gains. This is timely in the current policy window where the idea of a 'facility counsellor' is being discussed, whose role may span a range of counselling duties including: adherence counselling for TB/HIV, chronic non-communicable physical diseases and mental health counselling. The integration of services is a key health systems focus[66] requiring economic evidence.[23] The integration of services at the primary care level[67] to manage communicable and non-communicable chronicity is a global challenge[20 22 68 69] and a key strategic aim of the Department of Health[70] for expanding access to mental health services and managing chronic diseases in South Africa.[27 70] Brief interventions for depression and alcohol use disorders lack a common outcome measure for use in CEA,[71] thus limiting comparison across interventions.[72] This study will provide unique cost-effectiveness evidence using the QALY as an outcome measure for such interventions in the South African context.[51] This will allow comparability of cost-effectiveness results[73] and inform value for money decision making.[48 74] This is of particular importance for South Africa as discussions around cost-effectiveness thresholds[74] are being increasingly presented in the health policy space in the prelude to National Health Insurance. This protocol also responds to the call to publish health economics protocols alongside trial protocols to reduce publication bias of randomised controlled trial results.[75]

Potential challenges to this study include recall bias due to the reliance on patient-reported health service use for our estimations of service utilisation. To limit recall bias, a 1-month recall period will be used for ambulatory services and 6-month recall for hospitalisations.[52] Another possible limitation is that only a few patients may be referred for mental healthcare for CMD in the TAU option so we may not fully capture referral costs thus potentially underestimating cost-effectiveness. Another potential limitation is that the short follow-up period of 12 months may not capture the extent of the benefit that patients and the health system could gain through the availability of this service. For example, the degree to which the socioeconomic benefits and the returns to mental health associated with such interventions[76] may not be fully evident by the 12-month end point. This is a common problem associated with trial-based economic evaluations.[77]

## FUTURE WORK

Further work associated with this economic evaluation includes the use of modelling techniques to estimate the long-term benefits of the intervention both at the patient and system level and applying cost-benefit techniques to strengthen the evidence base for the mental health investment case.

## CONCLUSION

Priority setting and decisions around the scale up of an integrated comprehensive primary care service need to be informed by micro-level and macro-level economic analyses of the investment requirements, impact and equity implications of delivering key services such as counselling. It is anticipated that the cost estimates, the CEA and budget impact analysis of this intervention will be a useful contribution to priority setting and policy making during the current reconfiguration of South Africa's health system.[78] It is hoped that the economic evaluation of project MIND will provide much needed evidence on the cost-effectiveness of integrating mental health services into primary care that could support a business case for investing in these services.

**Author affiliations**

[1]Health Economics Unit, University of Cape Town School of Public Health and Family Medicine, Cape Town, Western Cape, South Africa
[2]Alcohol and Drug Abuse Research Unit, South African Medical Research Council, Tygerburg, Western Cape, South Africa
[3]Department of Psychiatry and Mental Health, Alan J Flisher Centre for Public Mental Health, University of Cape Town, Cape Town, Western Cape, South Africa
[4]Institute of Psychiatry, Psychology and Neuroscience, Health Services and Population Research, King's College London, London, UK
[5]Desmond Tutu HIV Research Centre, University of Cape Town School of Public Health and Family Medicine, Observatory, Western Cape, South Africa
[6]Western Cape Department of Health, Cape Town, Western Cape, South Africa
[7]Health Economics Unit, School of Public Health and Family Medicine, University of Cape Town, Observatory, Western Cape, South Africa

**Acknowledgements** The authors would like to thank all WCDOH facilities, which have agreed to participate in the trial, and our Stakeholder Advisory Group for their contributions.

**Contributors** VM-M, SC and BM made substantial contributions to the conception and design of the study. The first draft of the manuscript was written by VM-M and critically revised by all authors for important intellectual content. All authors read and approved the final manuscript.

**Funding**   This work is supported by the joint funded initiatives of the British Medical Research Council, Wellcome Trust and DFID (MR/M014290/1). BM is supported by the South African Medical Research Council.

**Competing interests**   None declared.

**Patient consent for publication**   Not required.

**Ethics approval**   This study was approved by the Western Cape Department of Health (WCDoH) (WC2016_RP6_9), the South African Medical Research Council (EC 004-2/2015), the University of Cape Town (089/2015) and Oxford University (OxTREC 2–17).

**Provenance and peer review**   Not commissioned; externally peer reviewed.

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
