## [Reviewer comments · BMJ Open]

ARTICLE DETAILS

TITLE (PROVISIONAL)	Integrating a Brief Mental Health Intervention into Primary Care Services for HIV and Diabetes Patients in South Africa: Study Protocol for a Trial Based Economic Evaluation.
AUTHORS	Mutyambizi-Mafunda, Vimbayi; Myers, Bronwyn; Sorsdahl, Katherine; Lund, Crick; Naledi, Tracey; Cleary, S

VERSION 1 – REVIEW

REVIEWER	Sam Watson University of Warwick, UK
REVIEW RETURNED	01-Nov-2018

GENERAL COMMENTS	This protocol describes the method for the economic evaluation of an intervention to integrate mental health services into care services for HIV and diabetes. The protocol is clear, comprehensive, and presents a sound method. My comments are therefore minor and are mainly points of clarification: Can the authors clarify which costs fall into health service and societal perspectives? While the main trial protocol is cited in this protocol, it would be useful to have a little more detail here. In particular, can the authors provide some more detail on 'standard care' (MIND_TAU) – it's not wholly clear if it includes mental health services, and how these are delivered. Is MIND_TAU representative of typical care services in SA for HIV and diabetes? Can the authors also provide planned trial dates (perhaps in Figure 1)? Are baseline and endline assessments cross-sectional in the clusters or longitudinal? For the sensitivity analysis, it would be useful to say how they would be used – e.g. to generate ICER credible intervals or produce a CEAC. Additionally, how will the authors incorporate uncertainty about treatment effect size? Can the authors specify 'standard ranges' for discount rates? For statistical analysis of costs and effects, could the authors be a little more specific about the analyses. How will costs be compared between arms? For the linear regression, I'm assuming no covariates from the text, but how will baseline and endline be accounted for, if at all? (e.g. outcome is differences, or baseline is controlled for).
--

	A useful citation for cost-effectiveness of task shifting in HIV care and other contexts in LMICs (admittedly one of my own!) is https://doi.org/10.1186/s41256-018-0073-z
--	--

REVIEWER	Nazlee Siddiqui Australian College of Health Services Management, Tasmanian School of Business and Economics, University of Tasmania. Australia
REVIEW RETURNED	03-Nov-2018

GENERAL COMMENTS	Abstract The abstract needs to be amended. In the introduction, the authors need to make it more distinct whether the study is conducted from the perspective of the service providers or the health services. Another suggestion is to briefly clarify the two task sharing approaches of integrated services in the method section. The abstract mentions of “budget impact analysis”, but such analysis is not adequately described in the protocol. For example, the protocol did not adequately discuss plans to estimate the uptake of the proposed intervention and assess feasibility of budget in near future? The Protocol The last sentence on page 3 is defining the study objective. There is scope to rephrasing this sentence, bringing greater clarity to the study objective. For example, is the study objective of economic evaluation including assessment of cost or consequences or combination of both? A diagram to portray the different arms in the research design would help in communicating the critical points in the protocol. More information about the rationale to compare the interventions of MIND-DED, MIND-DES and base line scenario would have been useful. Reference to previous literature to justify the 1-year time frame of economic evaluation will make the method of the protocol more credible. A brief description of the community health workers and facility staff such as professional qualification, years of experience and gender would help to repeat the study. Further clarity is required in the reporting of applicable discount rate on page 6 (3%), page 7 (8%) and Table 2 (Planned sensitivity analysis). The authors could elaborate on the plan for data collection. For example, elaborate the ingredients and step-down method for collection of institutional costs. The sentence “This clinical trial is powered for clinical rather than economic outcomes [38]” is confusing and needs greater explanation. It would be good to provide more information about presenting ICER from societal perspective. What factors are considered in this perspective to make it different from health services perspective? The other issue to clarify whether ICER from health services perspective represent the scope of primary care services? That is, primary care services assessing feasibility of mental health intervention. Or should the ICER perspective match the scope of the “facility counsellor”, as discussed in the section of discussion. Clarity of this point is important for coherence of the protocol. Regarding analysis of ICER, providing specific examples of South African contexts and the cost-effectiveness thresholds would be very meaningful. The protocol will benefit from a specific section for analysis of budget impact. In absence of such section, the topic of “budget
---

	impact of equitably implementing the service” in the section of discussion is not very meaningful. The referencing style of the protocol needs more work. About 10% of the references are contemporary, belonging to publications in year 2017 and 2018. The protocol should be able to use higher share of contemporary literature, given the recent progress in the literature of integration of mental and physical care.
--	--

VERSION 1 – AUTHOR RESPONSE

Reviewer(s)' Comments to Author:

Reviewer: 1

Reviewer Name: Sam Watson

Institution and Country: University of Warwick, UK

Please state any competing interests or state 'None declared': None declared

Please leave your comments for the authors below

This protocol describes the method for the economic evaluation of an intervention to integrate mental health services into care services for HIV and diabetes. The protocol is clear, comprehensive, and presents a sound method. My comments are therefore minor and are mainly points of clarification:

3. Can the authors clarify which costs fall into health service and societal perspectives?

Agreed clarity is needed here, health services or health systems should read provider perspective. On page 5 we have replaced “health systems” with “provider” and detailed what these costs include:

- i. Full economic provider costs: of the mental health intervention; any cost offsets attributable to the intervention related to changes to the costs of HIV or Diabetes care at the primary care level; and changes to the costs of referral care (including tuberculosis, emergency department and inpatient care);

Societal includes both the provider and the patient perspective, for the patient costs we have also added more detail:

- ii. Patient costs: associated with the intervention which will include direct OOP payments to private health providers (consultations and medications etc.), travel and subsistence costs and productivity losses.

Under perspective on page 5, we have also added more detail:

In line with good practise recommendations, the analyses will also be presented from a societal perspective (including both provider and patient perspectives). While public sector primary health care is free at the point of use, patients incur time and travel costs when accessing care and may experience productivity losses; in addition, they may incur costs when using private sector health services. These costs will be collected to inform the patient perspective within the economic evaluation.

We have also edited Table 1 to reflect measurement and valuation of provider costs the heading of the table has also been edited by replacing “health services” with “provider”

	MEASUREMENT		VALUATION	
Cost component	Resources used	Quantities or utilisation	Valuation data	Allocation factor
INTERVENTION				
Capital costs				
Facility Readiness Workshops	Number of staff	Time from trial data	Cost of Employment converted to annual equivalent cost	DED/DES/TAU headcount
Intervention training	Number of staff	Time from trial data	Cost of Employment converted to annual equivalent cost	DED/DES/TAU headcount
Counselling room	Space	Square meters	Replacement value converted to annual equivalent cost	DED/DES/TAU headcount
Furniture & equipment	e.g. Tape recorders	Inventory	Replacement value converted to annual equivalent cost	DED/DES/TAU headcount
Vehicles	Vehicles	Inventory	Replacement value converted to annual equivalent cost	DED/DES/TAU headcount
Recurrent costs				
Counselling personnel	Number of staff (Lay counsellors; registered counsellors; clinical psychologist)	Time from time and motion tool	Cost of Employment X proportion of time	DED/DES/TAU headcount
Counselling supplies	Manuals, notebooks, pens	Number used - from trial data	Market value	DED/DES/TAU headcount
Non-intervention personnel	Number of staff	Time from facility data	Cost of Employment X proportion of time	Facility headcount
Utilities	Electricity, water, other utilities, phone, cleaning, transport, and stationery	Facility utilisation	Annual facility expenditure	Facility headcount
HIV/DIABETES SERVICE				
Capital costs				
Buildings	Space used for HIV/Diabetes service	Square meters	Replacement value converted to annual equivalent cost	HIV/Diabetes headcount
Equipment & furniture	Equipment and furniture used for HIV/Diabetes service	Inventory	Replacement value converted to annual equivalent cost	HIV/Diabetes headcount
Vehicles	Vehicles used for HIV/Diabetes service	Inventory	Replacement value converted to annual equivalent cost	HIV/Diabetes headcount
Recurrent costs				
HIV/Diabetes personnel	Number of staff and staff time spent on HIV/Diabetes service	Time from facility data	Cost of Employment X proportion of time	HIV/Diabetes headcount
Drugs	ARV/Diabetes medication	Treatment protocol	National tender prices	Direct allocation
Diagnostic tests	CD4 test, diabetes monitoring tests etc.	Treatment protocol	National Health Laboratory Service (NHLs) prices	Direct allocation
Non-HIV/Diabetes personnel	Managerial, cleaning and security staff	Facility utilisation	Facility expenditure	Facility headcount
Building operating and maintenance	Electricity, water, other utilities, phone, cleaning, transport, and stationery	Facility utilisation	Facility expenditure	Facility headcount
OTHER RELATED HEALTH CARE PROVIDERS				

Public clinic visit	Average cost per visit	Patient interviews	Facility expenditure	Facility headcount
Public hospital ER/OPD visit	Average cost per OPD visit	Patient Interviews	Facility expenditure	Facility IP days and OP headcount
Public hospital Inpatient day	Average cost per inpatient day	Patient Interviews	Facility expenditure	Facility IP days and OP headcount

Table 1: Measuring and valuing provider costs

Under Estimating costs for the providers perspective on page 6, we have also made some further editions for clarity on costs :

Estimating costs for the provider's perspective

Within economic evaluation, the appropriate scope of provider costs includes all costs incurred within the intervention and any changes in broader health system costs that can be attributable to the intervention. We have categorized these costs as intervention costs, HIV and Diabetes service costs and other related provider costs. As shown in Table 1, our approach to estimating these costs entails the measurement of quantities of resources that are utilized, and multiplying these quantities by the value (or unit cost) of each resource. These separate steps of measurement and valuation are described below.

Measurement of resource use

For the intervention costs an ingredients approach will be used to estimate resources. Routine data linked to intervention protocols will be collected and used to assess resources directly consumed in the provision of the intervention, including supplies, manuals, and patient education materials.

4.While the main trial protocol is cited in this protocol, it would be useful to have a little more detail here. In particular, can the authors provide some more detail on 'standard care' (MIND_TAU) – it's not wholly clear if it includes mental health services, and how these are delivered. Is MIND_TAU representative of typical care services in SA for HIV and diabetes?

Yes, more care was needed here to explain standard care, which refers to standard of care for CMDs, on page 4 under Trial Design we have made the following insertion for clarity:

Treatment as usual (TAU) is representative of typical primary care level services for CMD in the South African public health service facilities, which is typically limited to referrals [25],[26].

Under Intervention and comparator on page 4 the following insertion was made:

In them TAU arm, the standard package of care will be provided to patients who are suspected of having mental health problems. In general, patients using the HIV or diabetes service are asked by a nurse or doctor attending to their care about their mental well-being, life stressors, and use of alcohol or other substances. Patients are provided with advice to change their lifestyles. Where the health care provider deems it necessary, patients are referred to a mental health nurse for further assessment or screening. The patient may also be referred to a social worker who may refer them to NGOs who provide counselling and support services [24].

Further reference to the TAU costs was also made on page 6:

The TMT will enable an analysis of differences in counsellor time usage in the different intervention modalities, indicating opportunities for scale and scope efficiencies. It is anticipated that the costs associated with the TAU option will be minimal and any referral costs will be captured through patient self-report of services they have used (see next paragraph).

And finally the potential impact on cost-effectiveness of the limited referrals in the TAU arm is acknowledged on page 11:

Another possible limitation is that only a few patients may be referred for mental health care for CMD in the TAU option so we may not fully capture referral costs thus potentially underestimating cost-effectiveness.

5. Can the authors also provide planned trial dates (perhaps in Figure 1)? Are baseline and endline assessments cross-sectional in the clusters or longitudinal?

Specifying trial dates is indeed helpful as is specification of the longitudinal nature of assessments, these editions have been added to the text under Outcome measures on page 5.

All outcomes are measured longitudinally at baseline, and at 6 and 12 month follow-up assessments, recruitment on the trial started in 2017 and will end in February 2019, and final outcome assessments will end in 2020 (see Figure 1).

Analysis	Assessment/ Economic Evaluation Outcomes	Measurement instrument OR Clinical Parameter	Assessment Timing (Trial recruitment: starts 2017- ends 2019)			
			Baseline	6 months	1 Year	
		Intervention start				Intervention assessment end
	1° Outcomes					
	Harmful alcohol use (Self-report interview)	AUDIT	✓	✓	✓	
	Depression (Self-report interview)	CES-D	✓	✓	✓	
	2° Outcomes					
CEA	Diabetes treatment failure risk	HbA1c	✓	✓	✓	
		BMI				
	HIV treatment failure risk	HIV-1 RNA (HIV-viral load)	✓	✓	✓	
	Chronic Disease Medication Adherence	Visual analogue Scale and CASE adherence Index	✓	✓	✓	
CUA	Multi-attribute Outcomes					
	QALY	EQ-5D	✓	✓	✓	

Figure 1 Economic Evaluation: Analyses, Outcomes, Measurement and Assessment Timing

6. For the sensitivity analysis, it would be useful to say how they would be used – e.g. to generate ICER credible intervals or produce a CEAC.

Indeed, the sensitivity analysis will be used to produce a CEAC as detailed in the following edition on page 9:

Probabilistic sensitivity analysis (which allows application of confidence intervals to point estimates) will be used for sensitivity analyses of all relevant individual level variables. The sensitivity analyses will be used to produce a cost-effectiveness acceptability curve (CEAC).

7. Additionally, how will the authors incorporate uncertainty about treatment effect size? Can the authors specify 'standard ranges' for discount rates? For statistical analysis of costs and effects, could the authors be a little more specific about the analyses. How will costs be compared between arms? For the linear regression, I'm assuming no covariates from the text, but how will baseline and endline be accounted for, if at all? (e.g. outcome is differences, or baseline is controlled for).

Thank-you, this has been addressed in table 2 providing more detail on the sensitivity analyses

Simple sensitivity analysis	Base-Case	Range
Discount rate	3%	0%-10%
QALYs: HRQoL weights	EQ-5D valued using UK TTO	EQ-5D valued using Zimbabwe TTO
Unit costs	Primary data and data from published sources and official statistics for South Africa	Varied within plausible ranges as determined from a literature review of South African cost studies
Missing data	Multiple imputation	Complete case analysis
Probabilistic sensitivity analysis		
Intervention utilization data	Mean value	95% uncertainty interval
Referral service utilization data	Mean value	95% uncertainty interval
EQ-5D/CESD/AUDIT data	Mean value	95% uncertainty interval
Clinical outcomes (HbA1c, HIV-1 RNA)	Mean value	95% uncertainty interval

Table 2: Planned sensitivity analyses

8. A useful citation for cost-effectiveness of task shifting in HIV care and other contexts in LMICs (admittedly one of my own!) is <https://doi.org/10.1186/s41256-018-0073-z>

Thank-you very much! This has been incorporated into the introduction and discussion.

Reviewer: 2

Reviewer Name: Nazlee Siddiqui

Institution and Country: Australian College of Health Services Management, Tasmanian School of Business and Economics, University of Tasmania. Australia

Please state any competing interests or state 'None declared': None declared.

Please leave your comments for the authors below

Abstract

The abstract needs to be amended.

9. In the introduction, the authors need to make it more distinct whether the study is conducted from the perspective of the service providers or the health services.

Thank-you, the need for clarity here is appreciated, we have amended the introduction as follows:

This work presents a protocol for a cost-effectiveness and cost-utility analysis from a societal (i.e. provider and patient perspective) of a task-sharing strategy for delivering a brief mental health counselling intervention to patients with comorbid HIV or Diabetes in South Africa.

10. Another suggestion is to briefly clarify the two task sharing approaches of integrated services in the method section.

Clarity is indeed required: the difference between the 2 alternatives is that in the MIND_DED the CHW will be added to the facility staff complement and will only deliver the new counselling service, whereas in the MIND_DES a CHW already working in the facility will be chosen to deliver the new counselling service on top of their existing duties. Italics have been used to emphasise the difference on page 4.

The two models will be the dedicated and designated models of care. In the dedicated approach (MIND_DED), community health workers (CHWs) will be hired and added to the facility staff complement and will dedicate their time to only delivering the new counselling service. In the designated approach (MIND_DES) CHWs already working in the facility will be designated to deliver the service in addition to their other chronic disease-related activities such as adherence counselling for HIV and health promotion.

11. The abstract mentions of "budget impact analysis", but such analysis is not adequately described in the protocol. For example, the protocol did not adequately discuss plans to estimate the uptake of the proposed intervention and assess feasibility of budget in near future?

Apologies for this omission, this has been addressed on page 9 by adding a Budget Impact Analysis section after the Sensitivity Analysis section as follows:

Budget Impact Analysis

Trial based estimates of intervention uptake and population based estimates of number in need will be used in budget impact approximations which will be conducted using a mathematical programming approach [52], [53]. These approximations will inform discussions on the fiscal implications of investing in primary care mental health services [54].

The Protocol

12. The last sentence on page 3 is defining the study objective. There is scope to rephrasing this sentence, bringing greater clarity to the study objective. For example, is the study objective of economic evaluation including assessment of cost or consequences or combination of both?

Agreed clarity would be helpful here, we have addressed this by the following edition on page 3:

This work presents a protocol for a cost-effectiveness and cost-utility analysis from a societal (i.e. provider and patient perspective) of a task-sharing strategy for delivering a brief mental health counselling intervention to patients with comorbid HIV or Diabetes in South Africa.

13. A diagram to portray the different arms in the research design would help in communicating the critical points in the protocol.

Thank-you for this suggestion – figure 2 has now been added, and reference to it in the text is found on page 8:

Figure 2 summarises the intervention arms, costs, and outcomes that will be included in this study.

Figure 2: Project MIND_ Intervention Arms, Costs, Outcomes & Economic Evaluation

14. More information about the rationale to compare the interventions of MIND-DED, MIND-DES and base line scenario would have been useful. Reference to previous literature to justify the 1-year time frame of economic evaluation will make the method of the protocol more credible.

Thank-you this comment is appreciated and on page 4 under Intervention and comparator reference is made to relevant literature informing the content of the underlying counselling intervention, its efficacy and time frame for assessment informing the evaluation time frame :

The underlying theory, content and evidence for the efficacy of the counselling approach for reducing symptoms of depression and hazardous /harmful alcohol use over the 1 year time frame have been described previously [29]–[31]

15. A brief description of the community health workers and facility staff such as professional qualification, years of experience and gender would help to repeat the study.

The need for clarity is acknowledged and the reader is referred to the trial protocol and this edition is made on page 4 under trial design:

Descriptions of the CHWs and registered psychological counsellors roles in the MIND-DED and MIND-DES models and their qualifications and skills levels are detailed in the trial protocol [24].

16. Further clarity is required in the reporting of applicable discount rate on page 6 (3%), page 7 (8%) and Table 2 (Planned sensitivity analysis).

Sorry for the confusion here. On page 5, we refer to the use of a discount rate to capture time preferences, and here we aim to use 3% in line with other published studies and recommendations from guidelines for economic evaluation.

On page 7 we are referring to the annuitization of capital costs. Here, the recommendation from guidelines for economic evaluation is to use an interest rate that reflects the long-term return on government bonds; in South Africa this interest rate is approximately 8%. We have edited the section on page 7 to improve clarity, as outlined below:

For capital items, costs will be estimated using the replacement value method [37]. In essence, the current replacement value of each item is calculated and annuitized using data on the estimated useful life of the item and an interest rate representing the opportunity cost component for capital, in order to estimate an equivalent annual cost. An estimated useful life of between 5-20 years for furniture and buildings will be applied as while an interest rate of 8% will be used as the annuitization factor based on the rate of return on government bonds in South Africa [37].

17. The authors could elaborate on the plan for data collection. For example, elaborate the ingredients and step-down method for collection of institutional costs.

Apologies for this omission, we elaborate on the ingredients method on page 6 and step down methodology on page 7 under Measurement of resource use as follows:

Page 6

Measurement of resource use

For the intervention costs an ingredients approach will be used to estimate resources. Routine data linked to intervention protocols will be collected and used to assess resources directly consumed in the provision of the intervention, including supplies, manuals, and patient education materials.

Page 7

For the HIV and Diabetes services, a step-down methodology will be used to establish resource use. Staff time will be determined through review of facility organograms and interviews with senior managers. In addition, utilisation of HIV and Diabetes medicine and diagnostics will be estimated by applying protocol-based facility utilisation statistics.

18. The sentence “This clinical trial is powered for clinical rather than economic outcomes [38]” is confusing and needs greater explanation.

Apologies, we agree, and the editions have been made on page 5 under Sample Size and Patient Population:

This clinical trial is powered to detect clinical outcomes, specifically reductions hazardous/ harmful alcohol use and risk of depression at 12-month follow-up rather than economic outcomes [24],[35].

19. It would be good to provide more information about presenting ICER from societal perspective. What factors are considered in this perspective to make it different from health services perspective? The other issue to clarify whether ICER from health services perspective represent the scope of primary care services? That is, primary care services assessing feasibility of mental health intervention. Or should the ICER perspective match the scope of the “facility counsellor”, as discussed in the section of discussion. Clarity of this point is important for coherence of the protocol.

The need for clarity here is appreciated, we have edited the manuscript to clearly show the costs for the provider perspective (page 6) and the patient perspective (page 7) and the ICER measurements (page 9) as follows:

Page 6

Estimating costs for the provider’s perspective

Within economic evaluation, the appropriate scope of provider costs includes all costs incurred within the intervention and any changes in broader health system costs that can be attributable to the intervention. We have categorized these costs as intervention costs, HIV and Diabetes service costs and other related provider costs. As shown in Table 1, our approach to estimating these costs entails the measurement of quantities of resources that are utilized, and multiplying these quantities by the value (or unit cost) of each resource. These separate steps of measurement and valuation are described below.

Page 7

Estimating costs for the patient perspective

Measurement of resource use

For the patient perspective, we will use a questionnaire at baseline, 6 and 12 months to estimate resource use. This will capture any user fees associated with public, private or NGO-provided health services as well as traditional and faith-based therapies. In addition, we will ask patients to estimate their time spent accessing health care services, as well as out of pocket payments for transport, subsistence and accommodation. Travel, subsistence and accommodation costs will relate to both the patient and their caregiver, if the patient is accompanied by a caregiver to any of these services.

Page 9

As the trial is focused on health systems strengthening for the delivery of mental health services, the decision rule applies to maximising health within the health care budget constraint. Therefore ICERs will be calculated from the provider perspective and , the cost per QALY gained will be compared to the cost-effectiveness thresholds (CETs) for LMIC settings [47]. If our intervention’s ICER is less than the chosen CET this will mean that diverting resources to the intervention will increase population health, and if the ICER is more than the CET the intervention is not cost effective. Patient costs will be collected and reported on separately. In addition, to satisfy the requirements of common guidelines [46] and to allow for some degree of comparability with other studies, the ICERs will also be presented for the societal perspective, i.e. including both provider and patient costs.

20. Regarding analysis of ICER, providing specific examples of South African contexts and the cost-effectiveness thresholds would be very meaningful.

Thank-you for this point, there are no CET for South Africa, we will make use of thresholds for LMICs , as per this insertion on page 9 under the section Determining Cost Effectiveness:

Therefore ICERs will be calculated from the provider perspective and , the cost per QALY gained will be compared to the cost-effectiveness thresholds (CETs) for LMIC settings [47]. If our intervention’s ICER is less than the chosen CET this will mean that diverting resources to the intervention will increase population health, and if the ICER is more than the CET the intervention is not cost effective. Patient costs will be collected and reported on separately. In addition, to satisfy the requirements of common guidelines [46] and to allow for some degree of comparability with other studies, the ICERs will also be presented for the societal perspective, i.e. including both provider and patient costs.

21. The protocol will benefit from a specific section for analysis of budget impact. In absence of such

section, the topic of “budget impact of equitably implementing the service” in the section of discussion is not very meaningful.

Sorry, we have added the following insertion on page 9 after the sensitivity analysis section:

Budget Impact Analysis

Trial based estimates of intervention uptake and population based estimates of number in need will be used in budget impact approximations which will be conducted using a mathematical programming approach [52], [53]. These approximations will inform discussions on the fiscal implications of investing in primary care mental health services [54].

22. The referencing style of the protocol needs more work. About 10% of the references are contemporary, belonging to publications in year 2017 and 2018. The protocol should be able to use higher share of contemporary literature, given the recent progress in the literature of integration of mental and physical care.

Thank-you for pointing this out, the references in the manuscript have been updated.

VERSION 2 – REVIEW

REVIEWER	Sam Watson University of Warwick, UK
REVIEW RETURNED	14-Jan-2019

GENERAL COMMENTS	The authors have satisfactorily addressed my comments and I can recommend this article for publication
--

REVIEWER	Nazlee Siddiqui Australian College of Health Services Management, Tasmanian School of Business and Economics, University of Tasmania, Australia.
REVIEW RETURNED	09-Jan-2019

GENERAL COMMENTS	The revised protocol has improved credibility of the manuscript. I believe a final check of the whole manuscript will help addressing few very minor issues in the manuscript. I have listed some of these minor issues below:  • Some sentences could be better phrased for greater succinctness. An example in case is the following statement on page 3: This work presents a protocol for a cost-effectiveness and cost utility analysis from a societal (i.e. provider and patient perspective) of a task-sharing strategy for delivering a brief mental health counselling intervention to patients with comorbid HIV or Diabetes in South Africa. A probable rephrasing for consideration of the authors is: This work presents a protocol of a task-shared mental health counselling intervention to patients with comorbid HIV or Diabetes in South Africa, for cost-effectiveness and cost utility analysis from societal perspective (i.e. providers and patients). • Similarly, the statement “The objective of this prospective economic evaluation is to estimate for alternative interventions the:” on page 5 needs amendment. • A brief explanation of what is meant by “protocol-based facility utilisation statistics” (page 5) would have been good.
---

	 • In Table 1, intervention training is identified as “capital cost”. Usually training is an expense item of regular nature? • On page 11, there is duplication of words in the section of “Budget Impact Analysis”. In the same section, few sentences to explain how the mathematical program will address the budget impact of equitably implementing the service would have been good. • The abstract should be changed to reflect the revisions made in the manuscript. For example, the last sentence in the introduction component of the abstract should mention “testing the cost-effectiveness and cost utility of two tasks”.
--	---

VERSION 2 – AUTHOR RESPONSE

Reviewer: 2

Reviewer Name: Nazlee Siddiqui

Institution and Country: Australian College of Health Services Management, Tasmanian School of Business and Economics, University of Tasmania, Australia.

Please state any competing interests or state ‘None declared’: None

Please leave your comments for the authors below

The revised protocol has improved credibility of the manuscript. I believe a final check of the whole manuscript will help addressing few very minor issues in the manuscript. I have listed some of these minor issues below:

1. Some sentences could be better phrased for greater succinctness. An example in case is the following statement on page 3:

This work presents a protocol for a cost-effectiveness and cost utility analysis from a societal (i.e. provider and patient perspective) of a task-sharing strategy for delivering a brief mental health counselling intervention to patients with comorbid HIV or Diabetes in South Africa.

A probable rephrasing for consideration of the authors is:

This work presents a protocol of a task-shared mental health counselling intervention to patients with comorbid HIV or Diabetes in South Africa, for cost-effectiveness and cost utility analysis from societal perspective (i.e. providers and patients).

Agreed the text has been adjusted as follows:

This work presents a protocol of a cost-effectiveness and cost utility analysis from a societal perspective (i.e. providers and patients) of a task-shared mental health counselling intervention for patients with comorbid HIV or Diabetes in South Africa.

2. Similarly, the statement “The objective of this prospective economic evaluation is to estimate for alternative interventions the:” on page 5 needs amendment.

Agreed the text has been adjusted as follows:

The study will be a prospective economic evaluation. The objectives include estimating:

3. A brief explanation of what is meant by “protocol-based facility utilisation statistics” (page 5) would have been good.

Thank-you for highlighting this, for clarity the following change has been made:

In addition, utilisation of HIV and Diabetes medicine and diagnostics will be estimated by applying standard treatment guidelines/protocols for HIV and Diabetes treatment in primary care services to the utilisation statistics obtained from facility records (i.e. protocol-based facility utilisation statistics).

4. In Table 1, intervention training is identified as “capital cost”. Usually training is an expense item of regular nature?

Thank-you for highlighting this. The training here refers to the training of the counsellors. This is defined as a start-up cost and as such reported as a capital cost. We have edited the table for and the text for clarity and added a reference to the Global Health Cost Consortium (2017) which defines start-up costs as including the costs of training staff.

INTERVENTION						
Capital costs						
Facility Readiness Workshops	Number of staff	Time from trial data	Cost of converted equivalent cost	Employment to annual	DED/DES/TAU headcount	
Intervention training for counsellors	Number of counsellors	Time from trial data	Cost of converted equivalent cost	Employment to annual	DED/DES/TAU headcount	

The costs of intervention focused counsellor training and facility-level institutional strengthening and capacity development through organisational readiness workshops [41] will be included in the cost analysis as start-up costs [42] in order to capture the full economic costs of integrating the intervention into the primary care service for chronic disease patients [42].

5. On page 11, there is duplication of words in the section of “Budget Impact Analysis”.

Thank-you so much for this, the duplication has been removed.

Trial based estimates of intervention uptake and population based estimates of number in need of the intervention will be used in budget impact approximations

6. In the same section, few sentences to explain how the mathematical program will address the budget impact of equitably implementing the service would have been good.

Thank-you for this input, this section has been reworked in line with the reviewer’s recommendations

There is an increased demand from decision makers in both LMIC and HIC settings for estimates of the real fiscal implications of investing in health services within a defined budget and budgetary period [53], [54]. Trial based estimates of intervention uptake and population based estimates of number in need of the intervention will be used in budget impact approximations [55] which will be conducted using a mathematical programming approach [56], [57]. Mathematical programming provides a useful equity lens for informing policy decisions around the provision of care in high burden, high cost contexts [58]. It will involve analysing alternative implementation strategies with differing equity implications subject to a budget constraint. This approach aims to support policy making by applying principles of constrained maximisation in resource allocation [58]. The estimates from this study will inform policy-level discussions on the fiscal implications of investing in primary care mental health services.

7. The abstract should be changed to reflect the revisions made in the manuscript. For example, the last sentence in the introduction component of the abstract should mention “testing the cost-effectiveness and cost utility of two tasks”.

Thank-you, the suggested change has been made:

This paper describes the proposed economic evaluation of a health systems intervention testing the effectiveness, cost-effectiveness and cost-utility of two task-sharing approaches to integrating services for common mental disorders with HIV and Diabetes primary care services.